# The Metabolism Reprogramming of microRNA Let-7-Mediated Glycolysis Contributes to Autophagy and Tumor Progression

**DOI:** 10.3390/ijms23010113

**Published:** 2021-12-22

**Authors:** Chien-Hsiu Li, Chiao-Chun Liao

**Affiliations:** 1Genomics Research Center, Academia Sinica, Taipei 115, Taiwan; dicknivek@icloud.com; 2Department of Tropical Medicine, National Yang Ming Chiao Tung University, Taipei 112, Taiwan; 3Institute of Public Health and Department of Social Medicine, National Yang Ming Chiao Tung University, Taipei 112, Taiwan

**Keywords:** glycolysis, autophagy, cancer, *Let-7*, microRNA

## Abstract

Cancer is usually a result of abnormal glucose uptake and imbalanced nutrient metabolization. The dysregulation of glucose metabolism, which controls the processes of glycolysis, gives rise to various physiological defects. Autophagy is one of the metabolic-related cellular functions and involves not only energy regeneration but also tumorigenesis. The dysregulation of autophagy impacts on the imbalance of metabolic homeostasis and leads to a variety of disorders. In particular, the microRNA (miRNA) *Let-7* has been identified as related to glycolysis procedures such as tissue repair, stem cell-derived cardiomyocytes, and tumoral metastasis. In many cancers, the expression of glycolysis-related enzymes is correlated with *Let-7*, in which multiple enzymes are related to the regulation of the autophagy process. However, much recent research has not comprehensively investigated how *Let-7* participates in glycolytic reprogramming or its links to autophagic regulations, mainly in tumor progression. Through an integrated literature review and omics-related profiling correlation, this review provides the possible linkage of the *Let-7* network between glycolysis and autophagy, and its role in tumor progression.

## 1. Introduction

Cellular energy-related metabolisms involve complex regulation dynamic processes. The current understanding is that the uptake of glucose from the extracellular environment is a primary way for cells to acquire resources for sustaining energy. Intermediate glucose metabolism can be converted by diverse metabolites of lipids and amino acids to maintain cellular functions [1]. In addition, autophagy is recognized as a digesting process to engulf cellular compartments or damaged organelles for maintaining metabolic homeostasis while responding to multiple metabolic stresses [2]. Within such processes, necessary molecules can be recycled by degrading specific factors to adapt cell growth to a rigorous environment. The glucose metabolic networks regulated by glycolysis and autophagy have explained the fundamental nutrients dynamic for maintaining cell growth and survival. Among them, miRNA, a 18–25-nt single-stranded noncoding RNA, serves as an essential modulator involved in cellular metabolisms, conducting post-transcriptional modification by targeting to 3’UTR of specific mRNA [3].

*Let-7* is the first miRNA family identified as involved in multiple cellular and biological functions, including glucose metabolism and autophagy. The glucose metabolism is controlled by the miRNA family of *Let-7* directly [4], or regulated by an autophagy-associated glycogen recycling system [5,6]. The imbalance of *Let-7*-mediated processes of glucose metabolism has been found to contribute to disease progression, especially carcinogenesis. In addition, metabolic dysregulation, which causes excessive energy release for unlimited growth, has been a consequential risk for promoting cancer development. However, the crosstalk networks between autophagy and glucose metabolism—especially the linkage of *Let-7* miRNA that participates in carcinogenesis and various biological functions—are still obscure and need to be fully addressed.

In this review, the connection between the *Let-7* family, glycolysis, and autophagy in glucose metabolism is comprehensively dissected and discussed. Accordingly, we also highlight the potent molecules and pathways involving glycolysis and autophagy and provide information on *Let-7* family-associated linkage with disease progression, mainly in tumorigenesis. The interplay between *Let-7*, autophagy, and glucose metabolism is an aspect of disease progression that will provide extensive knowledge for developing alternative cancer treatment strategies by the regulation of cellular metabolism.

## 2. Involvement of *Let-7* in Glycolysis Reprogramming

*Let-7* was reported in 1990 and contributes to the embryonic development of *C. elegans*. The artificial manipulation of the expression of *Let-7* causes mortality during embryogenesis [7]. Interestingly, several cancer-associated molecules have been identified from embryonic development, including *Let-7*. The *Let-7* family has been classified by its consensus sequence [8] (Table 1). According to the literatures review, the *Let-7* family-related expression was associated with the patient’s prognosis (Table 2). Furthermore, numerous studies have indicated that the related expression of *Let-7* is lower in tumor cells, whereas an increased level of *Let-7* is able to suppress tumor malignancy, which indicates that *Let-7* may contribute to the suppression role in most types of tumors [9,10].

There are divergent theories about how carcinogenesis starts. The monosaccharide glucose is the primary nutritarian for cells. After a meal, insulin increases and stimulates cell response to process glucose metabolism. Once cells uptake glucose, they undergo a process of glycolysis to convert glucose to other intermediates via specific enzymes and generate cellular components, including lipids, amino acids, and energy for cell survival [11]. According to the concept of cancer energy uptake raised by Douglas Hanahan and Robert Weinberg, the dysregulation of metabolism contributes to cancer progression [12]. Studies have demonstrated that the glucose level might change the mitochondria respiration in cells by modulating the expression of the *Let-7* level [13]. Comprehensive miRNA profiling from 14 global population studies indicated that the top 1% of population-differentiated miRNA was associated with glucose/insulin metabolism and pathogenesis. *MiR-202*, as one of the *Let-7* family members, may contribute to cancer progression by regulating glucose metabolism [14]. Additionally, Serguienko et al. observed that *Let-7* is linked to the expression of Glucose-6-phosphate Dehydrogenase (G6PD), Inosine-5’-monophosphate dehydrogenase 2 (IMPDH2), Fatty Acid Synthase (FASN), stearoyl-CoA desaturase, and Aminoadipate-Semialdehyde Dehydrogenase-Phosphopantetheinyl Transferase (AASDHPPT) from a comparable transcriptome analysis [15]. We herein describe the molecular mechanism of *Let-7*-mediated glucose metabolism and *Let-7*-associated metabolic reprogramming impacts in tumor plasticity (Figure 1).

The diagram summarizes the current participation of the Let-7 family in the regulation of glucose metabolism and autophagy. The direct (marked blue) and indirect (marked red) interrelationship between glycolysis- and autophagy-related pathway were highlighted according to the simulated model. Molecules and factors involved in the biogenesis of Let-7 and the speculation of its interaction with glucose metabolism and autophagic degradation were also illustrated. Possible molecules that regulate the Let-7 homeostasis in between non-carbohydrate metabolism and autophagy processes were indicated.

### 2.1. GLUT12

As reported, tumor cells tend to switch carbohydrate metabolism by changing the method of glucose uptake and the modulation of glucose transporters. According to the findings of Shi et al., the expression of glucose transporters (GLUT) 12 is associated to the poor prognosis in triple-negative breast tumors and negatively to the expression of *Let-7a*. Experimental assays have further demonstrated that *Let-7a* modulates GLUT12-mediated tumor growth and motility by targeting 3’UTR of GLUT12. Additionally, *Let-7a*-suppressed mitochondria respiration can be rescued by overexpression of GLUT12 [23].

### 2.2. HK2

Jian et al. analyzed the *Let-7* cluster function in terms of immunoglobulin production in B cells through a serial mouse genetic manipulation model. They found that *Let-7a/d/f* regulates glycolysis by inhibiting Hexokinase-2 (HK2) and modulates glutamine uptake by suppressing the glutamine transporter (Solute Carrier Family 1 Member 5, Slc1a5) axis, as well as glutaminase (Gls) via c-Myc, to restrict the tricarboxylic acid cycle (TCA cycle), consequently changing the ability of B cells to produce specific IgM [61]. As classified in the *Let-7* family, the *miR-98* has been identified as having lower expression in colon cancer tissue compared to normal tissue. The conducted two-way model revealed that *miR-98* was participated to cells’ proliferation ability. Consistent with *Let-7a* functions, the production of lactate and mitochondria respiration can be inhibited by targeting 3’UTR of HK2 to inhibit tumor growth [62]. A similar phenomenon was observed, *MiR-98* targets the 3’UTR of MAP kinase phosphatase 1 (MKP1) to modulate mitochondria respiration in non-small-cell lung cancer, although the evidence of how MKP1 regulates glycolysis remains insufficient [63]. *MiR-202*, another *Let-7* member, has also been demonstrated by Deng et al.’s group to target HK2 by a similar approach. They revealed that the expression of *miR-202* was decreased in chronic myeloid leukemia cells, and an increase of the *miR-202* level inhibited cell proliferation ability. In imatinib-resistant cells with high glycolysis activity, overexpression of *miR-202* increased imatinib sensitivity by restricting the expression of HK2, GLUT1, and lactate dehydrogenase A (LDHA) [64].

The relatively low level of *Let-7* expressed in a variety of cancers may explain the role of those *Let-7* downstream molecules in the regulation of drug resistance. Inhibition of *Let-7* has usually been observed in drug-resistant cells [65,66]. Li et al. found that *Let-7i* was decreased in cisplatin-resistant lung cancer cells [67], consistent with the related expression of *miR-202*, which was negatively correlated with imatinib-resistant ability [64]. Several essential enzymes of glycolysis—including HK2, pyruvate kinase M1/2 (PKM1/2), GLUT1, and LDHA—were increased in the cisplatin-based resistant cell model [65]. The events of *miR-202* targeting HK2 were consistently reported in hepatocellular carcinoma and pancreatic cancer [43,59].

### 2.3. ALDOC

Common metabolic-related diseases such as diabetes have been reported to be correlated with cancer progression. Studies have identified that the distribution of BCDIN3 Domain Containing RNA Methyltransferase (BCDIN3D), a RNA methylate in type II diabetes, is correlated with breast cancer prognosis. Knockdown BCDIN3D suppresses breast cancer growth by modulating mitochondrial respiration. RNA-seq and proteomics-based profiling showed that BCDIN3D-dominant mTOR signaling regulates glucose-related enzymes. Aldolase C (ALDOC) results in fructose 1,6 bisphosphate (F1,6BP) intermediate accumulation. Mechanistically, BCDIN3D regulates ALDOC via the *Let-7* family, including *Let-7b*, *Let-7d*, *Let-7e*, *Let-7f*, *Let-7g*, *Let-7i*, and *miR-98* [68]. ALDOB/C has been reported to be involved in fructolysis [69], indicating that *Let-7* may regulate the noncarbohydrate carbon substrates metabolism pathway to control mitochondria-related glucose metabolism.

### 2.4. PKM2

Increasing the level of *Let-7* accelerates mitochondrial activity by increasing the membrane potential, oxygen consumption rate, and extracellular acidification rate, which changes the method of glucose metabolism by disturbing the aerobic glycolysis (the Warburg effect) of cancer cells. Although an accelerating rate of glucose uptake and upregulated activity of pyruvate kinase M2 (PKM2) can be observed in *Let-7*-overexpressing cells, the Warburg effect is no longer the primary processes of glucose metabolism in cancer cells [15]. Instead of aerobic glycolysis, cells have started to use oxidative phosphorylation and have consequently increased the related level of reactive oxygen species (ROS) in the mitochondria. Accumulated ROS under switched metabolism makes cells increase their oxidative response in response to therapeutic drugs. In addition, it has been speculated that the morphology change of cells is related to ROS-mediated anaerobic glycolysis, which involves the epithelial-to-mesenchymal transition [15]. However, further investigations are required to find out whether *Let-7* serves as a key to switch oxidative phosphorylation to the Warburg effect and thereby promotes cell malignancy. Notably, the overexpression of PKM2 has been identified as positively correlated with malignancy and poor prognosis in breast cancer. The knockdown of PKM2 inhibited the proliferation of breast cancer [70,71], thus indicating that *Let-7*-modulated PKM2 activity might be associated with the conversion of the original glucose metabolism pathway into cancer-favoring metabolic processes and sustained tumor malignancy.

Yao et al. provided evidence that an increase in *Let-7a* inhibited the expression of PKM2 as well as GLUT1 and Phosphofructokinase-1 (PFK1) in breast cancer [72]. A similar phenomenon was observed in glioma by Luan et al., who found that *Let-7a* restricts PKM2-mediated aerobic glycolysis and cell growth. Mechanistically, *Let-7a* suppresses Myc by targeting its 3′UTR to block the downstream heterogeneous nuclear ribonucleoprotein A1 (HnRNPA1)/PKM2 axis. HnRNPA1 forms a terminal loop with *Let-7a* by inhibiting its essential biogenesis factor, Drosha, to maintain the level of *pri-Let-7a* in glioma [73]. Such a regulation circuit may explain why Let-7 expression is inhibited upon biogenesis during carcinogenesis, resulting in cells switching the oxidative phosphorylation into the Warburg effect. Despite the fact that the tumor suppressor role of *Let-7* has been demonstrated in breast cancer and glioma, with controversial responses in terms of mitochondria respiration [15,73], the evidence suggests that *Let-7* may regulate the diverse activity of glucose metabolism-related enzymes to control tumoral functions. Biologically, *Let-7a* suppresses the Myc/HnRNPA1/PKM axis, which further modulates tumor growth and motility [74]. The regulation of the *Let-7*/Myc axis has also occurred in B-cells, and this *Let-7*-mediated regulation has been demonstrated to regulate glycolysis and glutamine uptake [61].

### 2.5. Noncarbohydrate Metabolism Crosstalk

*Let-7* has been considered a terminal differentiation factor and participates in postnatal cardiac maturation by switching glycolysis and fatty acid oxidation under limited glucose resources through the *Let-7g*- and *Let-7i*-mediated PI3K/AKT/insulin axis [75]. Additionally, gluconeogenesis, as an inverse glucose metabolism pathway, has been observed to be regulated by *Let-7*. Recently, Methyltransferase 3, N6-Adenosine-Methyltransferase Complex Catalytic Subunit (METTL3) has been reported to be associated with a poor prognosis of hepatocellular carcinoma. A gene set enrichment analysis revealed that METTL3 is correlated with glycolysis and gluconeogenesis-related enzyme levels. Knockdown of METTL3 decreases glucose uptake, consequently suppressing lactate production by altering the Warburg effect. Within this route, mTOR signaling is also suppressed by the downregulation of METTL3 [76]. METTL3 has been demonstrated to be a *Let-7g*-specific target in breast cancer. *Let-7g*-mediated METTL3 has been found to be negatively regulated by HBXIP to promote tumor growth. It participates in the feedback loops of the HBXIP/METTL3 axis by N6-methyladenosine modification and regulates glucose metabolism in breast cancer [77]. Apart from this, a high level of polyamines has been reported to contribute to cancer progression. The crosstalk between polyamine and glucose metabolism can be suppressed by 2-deoxy-d-glucose treatments [78]. In colorectal cancer, treating cells with difluoromethylornithine suppresses polyamines levels—including putrescine, spermidine, and spermine—and induces *Let-7i* expression. However, decreased polyamines, resulting in a downstream eIF5A1/A2 and LIN28 axis, were not able to suppress *Let-7i* expression in neuroblastoma [10]. This is connected to the role of Myc, a downstream target of *Let-7* that regulates protein synthesis, glycolysis, and polyamine synthesis. Notably, eIF5A1 was associated with protein synthesis, and such a regulation may link the *Let-7*-mediated metabolism regulation circuit between glucose and noncarbohydrates.

### 2.6. Oxidative Stress

Several genes related to energy metabolism—including MT1X, MT2A, MT1G, MT1A, SOD2, TXNRD1, GSTM3, CTH, HMOX1, and FTH1—have been reported to be associated with oxidative stress and regulated by *Let-7a*. Under hypoxia, HIF-1α was activated, promoting glucose metabolism and increasing tumor stemness [79]. Glycolysis-related enzymes, such as ALDOA with nonenzymatic functions, positively regulate the HIF-1α/ALDOA feedback loops to increase the activation of MMP9 in lung cancer under hypoxia [80]. Knockdown of carbonic anhydrase IX, a factor upregulated under hypoxia conditions, transactivates *Let-7d*, *Let-7c*, and *Let-7f* to suppress LIN28 as well as pyruvate dehydrogenase kinase 1 (PDK1) expression, leading to decreased tumor stemness activity [81].

### 2.7. Stemness Activity

Cai et al. identified that the stemness of breast cancer was controlled by the Wnt/β-catenin-mediated repression of the *Let-7* family. The decreased level of *Let-7* was regulated through the activation of Lin28. The involvement of the *Let-7*/Lin28 axis in the Wnt/β-catenin pathway was primarily found in breast cancer [82]. Their group further identified Chibby as a primary transcription factor of Wnt signaling, negatively regulating the expression of β-catenin in nasopharyngeal carcinoma. The clinical relevance of the expression level between β-catenin and Chibby in nasopharyngeal carcinoma was negatively correlated. Increased Chibby induces cells to switch the Warburg effect into oxidative phosphorylation to change ATP production, initiating oxygen consumption and lactate release. Mechanistically, Chibby suppresses Wnt/β-catenin/PDK1/Lin28/Let-7g to control tumor proliferation [83].

### 2.8. Compound-Related Regulators

Recently, several reports have revealed that *Let-7*-mediated glucose metabolism could be affected by specific compounds. Alharris et al. found that, by treating cells with phytocannabinoid-related molecules, cannabidiol can suppress *Let-7a* to modulate metabolism pathways and related downstream effectors, including GAS7 and CASP3, subsequently inducing apoptosis to suppress neuroblastoma progression [84]. Dichloroacetate was designed to change pyruvate dehydrogenase activity, which leads glucose metabolic intermediates into the TCA cycle to execute oxidative phosphorylation in mitochondria. Treating cancer cells with dichloroacetate decreases glucose metabolism and cell viability. Interestingly, it was observed that *Let-7a* and *Let-7c* could be transactivated by dichloroacetate in the breast cancer cell line MDA-MB-231 [85]. Overexpression of *Let-7* mimics the effects of dichloroacetate treatment on the induction of Bax/P53 cascade, increasing the expression of proliferator-activated receptor gamma co-activator 1 alpha (PGC1α) and Mitofusin 2 (MFN2) to accelerate mitochondria fusion and the release of ROS from oxidative phosphorylation, leading to apoptosis [85]. Although the effect of PGC1α and MFN2 on the Warburg effect remains unclear, these results suggest that *Let-7* could regulate glucose metabolism and tumor cells could switch glucose metabolism into the Warburg effect in the way of this *Let-7*-associated modification. In addition, *Let-7* could affect mitochondria stability by oxidative phosphorylation. RNA-seq profiling of rectal carcinoma from Chen et al. showed that *Let-7e* was correlated with the expression of peroxisome proliferator-activated receptor coactivator 1 alpha (PGC-1alpha) and mitochondrial biogenesis molecules, and may contribute to liver metastasis [86]. Similar observations were described by Xu et al., who described that ATP5A1 and ATP5B increased while *Let-7f* decreased in glioblastoma. A simulated analysis showed that *Let-7f* regulates oxidative photophosphorylation through the regulation of ATP5B [87].

## 3. *Let-7*-Mediated Autophagy Participates in Glucose Metabolism and Cancer Progression

### 3.1. Let-7 and Autophagy

In lung cancer, *Let-7* targets IGF-1R to induce autophagy and blocks the function of BCL2L1/BCL2/PI3K complex to induce apoptosis and pyroptosis and inhibit cell motility [88]. *Let-7a* targets Rictor’s mTORC2 component, inhibiting AKT/mTORC1 signaling to activate autophagy in gastric cancer [89]. Similar regulation can be observed in human placental trophoblasts, in which the expression of *Let-7b* was correlated with cell growth and motility. The *Let-7b*-mediated TGFBR1/ERK/IL-6/TNF-α cascade triggers not only apoptosis but also autophagy. Such regulation may contribute to pre-eclampsia during pregnancy [90]. In glioma, the downregulation of STAT3 was mediated by *Let-7a*, *Let-7d*, and *Let-7f*. Upregulation of *Let-7* suppressed the expression of STAT3, resulting in the inhibition of cell proliferation and induction of autophagy and apoptosis [91]. Liang et al. identified that a set of the *Let-7* family was downregulated in hepatocellular carcinoma, with different clinical correlations under a genetic profiling analysis. The expression of *Let-7b* and *Let-7c* had a better prognosis; *Let-7e* had a poor prognosis instead. Among them, *Let-7e* has been demonstrated to promote tumor growth by suppressing autophagy and apoptosis [92]. A similar strategy was used in cholangiocarcinoma. Clinical evidence showed that the expression of NUAK1 was negatively correlated with *Let-7a*. NUAK1-mediated cholangiocarcinoma cell motility can be suppressed by increasing *Let-7a*. In turn, the overexpression of *Let-7a* inhibited NUAK1-mediated tumor malignancy by the induction of autophagy [93]. Additionally, *Let-7* can be regulated by LncRNA H19 and LIN28 in breast cancer. The expression of long non-coding RNA (lncRNA) H19 and LIN28 was correlated with breast cancer’s poor prognosis and metastasis ability. Overexpression of H19 and LIN28 increases the expression of several autophagy-related ATG markers as well as its puncta structure formation. Downregulation of *Let-7* increased the transcript activity of several EMT-related genes—including Slug, Zeb1, Twist, Snail, β-catenin, and HMGA2—to modulate the metastasis of breast cancer [94]. Another lncRNA MIR99AHG, as well as its *Let-7c*-associated cluster, were reported to have decreased expression in lung cancer. MIR99AHG increased *Let-7c*, subsequently promoting autophagy via targeting mTOR, an autophagy suppressor of nucleation, and ANXA2, a negative regulator of elongation, to suppress the growth and motility of lung adenocarcinoma [95]. In view of the controversial role of autophagy in a variety of cancers, the regulation of *Let-7*-mediated autophagy in tumor progression could be complicated—and condition-, environment-, and tissue-specific.

### 3.2. Autophagy Activators

Several components have been identified as triggering *Let-7*-mediated autophagy in cancer cells. Treating cells with recombinant capsid protein viral particle 1 (rVP1) induces autophagy to regulate the motility of macrophages [96] and ovarian cancer cells [97]. In ovarian cancer, autophagy—activated by either a canonical or a rVP1-mediated noncanonical pathway—maintains the homeostasis of the *Let-7* level through SQSTM1-mediated degradation of Dicer/AGO2 inhibition of cell migration [97]. In lymphosarcoma, the expression of *Let-7g*/CTSB may be suppressed by ribonuclease binase to participate in apoptosis and autophagy [98].

### 3.3. Drug Resistance

In gastric cancer, the expression of *miR-202* can be restricted by lncRNA MALAT1, resulting in the activation of autophagy, increased tumor malignancy, and an enhanced drug-resistant ability [99]. In agreement with other reports, Yang et al. showed that paclitaxel-based drug-resistant breast cancer cells express a high level of CircRNA ABCB10 and autophagy, which are correlated with clinical paclitaxel-sensitive or resistant data and negatively associated to *Let-7a*. Mechanistically, the *Let-7a*/DUSP7 axis is a downstream effector of Circ-ABCB10 resistant to paclitaxel treatment. Knockdown of Circ-ABCB10 not only increases sensitivity to paclitaxel but also decreases tumor weight [100]. Similar regulation was observed in a cisplatin-based resistance model of A549 with a high level of DICER. Overexpression of DICER induces autophagy processes and increased tumor growth and motility, in which DICER-mediated suppression of *Let-7i* and the PI3K/AKT/mTOR axis contributes to the autophagy activity [67]. In medulloblastoma, inhibited autophagy was found to promote tumor resistance upon cisplatin treatment. The level of *Let-7f* in cells was insufficient to repress HMGB1 and led to autophagy-mediated drug resistance. Overexpression of *Let-7f* could attenuate cisplatin’s drug resistance and induce apoptosis in medulloblastoma cells [101].

### 3.4. Let-7-Mediated Autophagy in Glucose Metabolism

Recently, *Let-7*-mediated autophagy has been described as participating in glucose metabolism events. For example, Duan et al. observed that *Let-7* targeted BCL-xL to induce autophagic cell death in lung cancer, indicating that *Let-7* regulates mitochondria-related autophagy (mitophagy) to regulate metabolism-related events, and BCL-xL with non-apoptotic functions to induce cell death [102]. However, the underlying mechanism of *Let-7*-mediated autophagy in glucose metabolism that contributes to cell stress and death needs to be further elucidated. According to the above reports, several links may support the correlation between *L**et-7*, autophagy, and glucose metabolism. In turn, Lai et al. found that—in a hypoxic environment—HIF-1α can interact with DICER to regulate miRNA processing in diverse cancer types, including colon, breast, liver, lung, and prostate cancer [103]. HIF-1α changed the glycolysis-related enzyme PDK1 level and induced autophagy-mediated proteolysis by interacting with Parkin/p62 to possess DICER, which decreased *Let-7* biogenesis. Overexpression of HIF-1α reduced the levels of *Let-7a*, *Let-7b*, and *Let-7d* as well as its complement downstream target LIN41 and Aurora B to promote tumor metastasis [103]. However, how glycolysis participates in DICER ubiquitination and related autophagy processes has not yet been well explained. So far, Lai et al.’s study provides a possible reason for why the *Let-7* level being downregulated under hypoxia is an important factor contributing to tumor microenvironment reprogramming and providing tumor cells with escape from immune surveillance. Recently, bone marrow-derived human mesenchymal stem cells (hMSCs) have been observed to have anticancer activity. Egea et al. found that *Let-7f* can be transactivated under hypoxia to induce autophagy in hMSCs and promote migration in tumor cells [104]. *Let-7f* can be regulated by TGF-β, TNF-α, IL-1β, and SDF-1α to modulate CXCR4 and MMP-9 expression and drive chemotactic invasion. Interestingly, hMSCs have been observed to transport *Let-7f* by exosome secretion to inhibit the growth and motility of breast cancer; such events can be reversed with the *Let-7f* inhibitor [104].

### 3.5. mTOR-Dependent Autophagy and Glucose Metabolism

Several studies have found that *Let-7* mediates glucose metabolism through the regulation of mammalian target of rapamycin (mTOR) [67,68,76,89]. It is also a well-known negative regulator of autophagy. Notably, human growth hormone receptors (GHR) have played an essential role in glucose metabolism and are linked to mTOR activity. A murine model revealed that, under limited nutrients, growth hormone maintained the cellular glucose level through gluconeogenesis, accompanied by the induction of autophagy [105]. In addition, GHR has been identified to contribute to breast and prostate cancer malignancy [106,107]. Elzein et al. reported that GHR is the target of *miR-202*. Increased *miR-202* suppresses the expression of GHR in MCF and LNCaP cells [108]. Additionally, it has been reported that PKM2 and mTOR expression is downregulated under glucose restriction in breast cancer, which reverses the Warburg effect of cells [109]. Strikingly, these molecules were all be *Let-7* downstream effectors. Such regulation may explain how *Let-7* mediates autophagy and glucose metabolism to regulate cancer cell progression (Figure 1).

## 4. Possible Connections between *Let-7*-Mediated Glycolysis and Autophagy in Cancer Progression

According to the literature review, we separately discussed the link of the *Let-7* family to glucose metabolism and autophagy. The intermediates of noncarbohydrate metabolisms, such as amino acids and lipids, are also integrated into glucose metabolism and regulated by *Let-7* [61,75]. Our aim in this section is to profile a more comprehensive linkage between autophagy and glucose metabolism via *Let-7* regulation. We conducted a molecular regulation network simulation, including the upstream regulators and downstream effectors of the *Let-7* family. Table 3 lists the upstream regulators and downstream effectors of the *Let-7* family involved in autophagy and metabolism-related biological events. Moreover, according to the literature review, the *Let-7* family may be involved in glycolysis- and autophagy-related tumorigenesis, so simulated results of relative upstream regulators/downstream effectors are briefly displayed (Figure 1). Possible effects/mediators/molecules of the relationship between glucose metabolism and autophagy, and connecting networks, are also discussed.

### 4.1. Upstream Regulators

#### 4.1.1. LIN28

Several *Let-7* upstream regulators have been identified. The most well-known factor is LIN28. Zhou et al. observed that LIN28B affects diverse metabolism-related gene ontology, including the metabolism of cellular amino acids, oxoacids, organic acids, and carboxylic acid. LIN28B regulates IGF2BP1 via *Let-7a* or *Let-7b* to affect acute myeloid leukemia cell proliferation [110]. However, no direct evidence clarifies the link between *Let-7* and metabolic processes. Similar observations were described by Ackermann, who showed that the balance between *Let-7* and LIN28B can be controlled by C/EBPβ-LIP. Using mouse embryonic fibroblasts as a cancer metabolic reprogramming mimic model, their study indicated that C/EBPβ-LIP changed the mitochondrial respiration and served as a transcription factor to regulate the *Let-7* family—including *Let-7a*, *Let-7b*, *Let-7c*, *Let-7d*, *Let-7f*, *Let-7g*, and *Let-7i*—consequently modulating cell glycolysis, proliferation, tissue regeneration, and carcinogenesis [111].

The simulation of upstream regulators of *Let-7*—including AKT, AP1, CREB, E2F1, FOXO1, FOXO3, HIF-1α, Myc, and NF-κB—contributed to autophagy-related biological features. When cells are stimulated with toxins, metabolic stresses, ischemia, trauma, and inflammation trigger metabolic reprogramming in the mitochondria and ER, causing DNA damage and activating autophagy [112].

#### 4.1.2. AKT

Usually, with adequate nutrition, autophagy activity is suppressed by PI3K/AKT/mTOR signaling, which blocks the ULK complex (ULK1/2, FIP200, and ATG13) in autophagy initiation. Once in starvation conditions, autophagy is activated inversely [113]. In gastric cancer, AKT is activated by NEK2 to regulate cell proliferation. Phosphorylated AKT promotes cell switching to aerobic glycolysis, accompanied by increased glucose uptake and lactate production. Notably, increased NEK2 suppresses autophagy through the AKT/mTOR axis, and, by treating AKT-specific inhibitors, induces autophagy and reverses mitochondrial respiration by inhibiting GLUT1, PKM2, and HIF-1α [114]. A similar observation has been made in prostate cancer. AKT activity is inhibited by FGF21 and blocks mTOR signaling to drive autophagy [115]. AKT has been demonstrated to regulate *Let-7* to change glucose metabolism. In type II diabetes patients, the *Let-7* level is controlled by insulin/PI3K/AKT to govern glucose uptake in muscle cells [116]. This relationship indicates that the tumor suppressor *Let-7* expression could be dysregulated and may influence autophagy in the regulation of cell death or survival. These studies explain how diabetes is linked to cancer progression within *Let-7*/glycolysis/autophagy regulation.

#### 4.1.3. NF-κB

As a downstream factor of AKT, NF-κB has been demonstrated to regulate the *Let-7* family—including *Let-7a*, *Let-7b*, *Let-7f*, and *Let-7g*—by inhibiting LIN28B in castration-resistant prostate cancer [117]. However, no direct evidence shows that NF-κB-mediated *Let-7* participates in autophagy processes. Liang et al. observed that treating cells with galangin inhibits NF-κB signaling from activating autophagy in gastric cancer [118].

#### 4.1.4. FOXO

FOXO is associated with glucose and lipid metabolic processes [119]. The activity of FOXO can be regulated by insulin/AKT, and then translocates into the nucleus as a transcription factor [120]. In cancer, FOXO1 was activated by ROS to regulate autophagy and mitochondrial oxidative metabolism [121,122]. In breast cancer, the level of FOXO3 can be induced by rapamycin, AZD3463, and AZD-RAPA, accompanied by autophagy formation [123]. Hopkins’ group reported that FOXO3 could be regulated by peroxidase peroxiredoxin 1 to control *Let-7b* and *Let-7c*, affecting cell migration ability. Notably, the activity of FOXO3 is partially regulated by AKT [124].

#### 4.1.5. Myc

Myc is the *Let-7* downstream target, and Myc can regulate *Let-7* by LIN28 [125]. Interestingly, treating cells with aristolochic acid I can activate Myc and NF-κB to control the LIN28B/*Let-7b* axis, activating FOXO1 to promote tumorigenesis and resistance to apoptosis [126]. However, how AP1, CREB, and E2F1 regulate *Let-7* has not been reported.

### 4.2. Glycolysis

The downstream molecules/regulators of *Let-7* family are discussed in this section. ALDOC and phosphoglycerate kinase (PGK1) have been identified as possible downstream targets of *Let-7* (Table 3). Aldolases are an essential enzyme of gluconeogenesis in regulation of converting Fructose 1,6-Bisphosphate (F1,6BP) into dihydroxyacetone phosphate (DHAP) and glyceraldehyde 3-phosphate (G3P). In breast cancer, ALDOC can be regulated by *Let-7f* and contributes to type II diabetes-mediated breast cancer [68]. PGK1 has been observed to be correlated with poor prognosis of glioblastoma [127] and has been demonstrated to bind directly to the Beclin1 and ATG14, two ATGs required for the autophagy process [127,128]. Mechanistically, PGK1 can phosphorylate Beclin1 and promote phagophore formation, resulting in the induction of autophagy and tumor malignancy. Notably, such regulation is involved in glutamine deprivation events [127]. Even though no direct evidence clarifies the link between *Let-7* and PGK1, as reported, PGK1 is a critical mediator for AKT/mTOR signaling, which may link to autophagy activity [129,130]. Together with the identified upstream regulators of *Let-7*, AKT may serve as a link between PGK1 and *Let-7*. This evidence links *Let-7*-mediated autophagy to glucose metabolism, and the crosstalk of amino acid metabolism mediated by ALDOC and PGK1. Another similar postulation can be raised for *Let-7* and PKG1. PGK1 and HIF-1α can form a regulatory circuit for controlling breast cancer metastasis [131]. Additionally, Myc serves as an upstream regulator of *Let-7* (Table 3), while PGK1 is a downstream target of Myc [132,133,134]. The *Let-7* might function as a linker for Myc to carry out post-transcriptional modification on PGK1, given the possible connection between PKG1 and *Let-7*.

### 4.3. TCA Cycle

Citrate synthase activates AKT, *Let-7*’s upstream regulator, to modulate metastatic progression of triple-negative breast cancer [135]. Berberine has been reported to decrease the expression of citrate synthase in mitochondria, triggering autophagy and apoptosis in glioblastoma and pancreatic cancer cells [136]. Malate dehydrogenase is a novel autophagy regulator that has been identified in pancreatic ductal adenocarcinoma. Activation of malate dehydrogenase maintains the ULK1 level and improves resistance to starvation and hypoxia conditions. Interestingly, the activation of autophagy can induce the expression of malate dehydrogenase [137]. However, how PGK1, citrate synthase, and malate dehydrogenase are regulated by *Let-7* has not been fully investigated. The simulated results are provided, and the potent *Let-7* downstream molecules including citrate synthase and malate dehydrogenase that may be involved in autophagy regulation are documented in Table 3.

### 4.4. Glutamine

Glutaminase is an essential enzyme that can convert glutamine to glutamate. In B-cells, *Let-7* regulates glutaminase by Myc, consequently blocking glutamate conversion to α-ketoglutarate (α-KG) during the TCA cycle, which is related to IgM production in B-cells [61]. In cancer, the glutaminase level is correlated with colorectal tumor progression. Decreased glutaminase induces oxidative stress and inhibits autophagy formation by suppressing tumor growth and motility [138]. Mukha and his group identified that the glutamine level is correlated with radiosensitivity of prostate cancer. A global gene expression profiling metabolism signature shows radioresistant cells with high-level glutamine and α-KG in radioresistant PC-3 and DU145. Block supplies of glutamine or glutaminase induce radiosensitivity and trigger the autophagy progress [139]. Interestingly, α-KG has been observed to contribute to tumor stemness activity through JMJC-mediated histone demethylation [140]. A similar study, reported by Xia et al., showed that genetic modification, such as KRAS mutation, has different glucose and glutamine metabolic profiles. Treating cells with a glutaminase inhibitor can change the mitochondrial membrane potential to induce ROS production while decreasing AKT activity, and leads to activation of autophagy. A combination KRAS and glutaminase inhibitor had a synergistic effect on antitumor progression [141]. Mitochondrial damage and oxidative stress can trigger autophagy to remove damaged organelles and reduce oxidative pressure to decrease cytotoxicity. Selenium has been demonstrated to inhibit glutaminase and promote ROS levels to induce autophagy and apoptosis in lung cancer [142,143]. To sum up the evidence, *Let-7* inducing ROS levels or targeting the TCA cycle could be proposed as alternative strategies to control tumor progression in an autophagy-associated manner.

### 4.5. Arginine

In non-carbohydrate metabolism, arginase is an essential enzyme for arginine conversion to ornithine and glutamate by the urea cycle, which is crucial for the TCA cycle. Arginase is considered an essential marker in cancer [144,145,146] and may be targeted by *Let-7* (Table 3). Genetically modified mouse models show that the deletion of ATG5 or ATG7 induces an increase in the level of arginase 1 in serum, decreasing the circulating arginine and increasing the ornithine level to depress tumor growth [147]. Modified arginase 1 can reduce arginine levels and induce autophagy or apoptosis in colon cancer [148]. In the tumor microenvironment, arginase increased in Tim4^+^ Tumor-associated macrophages and was associated with mitochondrial respiration and autophagy (mitophagy). Genetically modified mouse models show that the deletion of FIP200 can inhibit autophagy and decrease the Tim4^+^ Tumor-associated macrophage population, consequently promoting the antitumor activity of T-cells [149].

### 4.6. Autophagy Processes

Stress-mediated 5′ adenosine monophosphate-activated protein kinase (AMPK) is known to activate the ULK complex, and its activity has been correlated with the glucose level [150]. Simulated results reveal that the *Let-7* family may contribute to AMPK expression. Recently, there has been evidence that the expression of *Let-7* is linked to glucose metabolism and insulin levels during pregnancy. Furthermore, *Let-7*-mediated AMPK contributes to liver metabolism [151]. Additionally, Zhong et al. revealed that one diabetes-related drug, metformin, increases AMPL activity and *Let-7* to regulate gene methylation in ARK2 and MCF-7 cells [152]. Moreover, simulated results show that the autophagy-related components might be regulated by *Let-7* (Table 3).

### 4.7. Oxidative Stress

Oxidative stress such as hypoxia is ubiquitous in tumors and the microenvironment. As an oxidative stress factors, HIF-1α-mediated *Let-7* participates in autophagy, and glucose metabolism has been reported [103]. In lung cancer, activated HIF1α is a transcription factor that transactivates ALDOA and promotes the Warburg effect, increasing lactate production under hypoxia. Interestingly, lactate, in turn, negatively regulates hypoxia-inducible factor 1 alpha and stabilizes HIF-1α, which forms a regulatory circuit to increase cell motility [80]. In addition, HIF-1α can modulate *Let-7a*, *Let-7b*, and *Let-7d* biogenesis by interacting with DICER to activate PDK1 [103]. The production of excess reactive oxygen species (ROS) is harmful for cells metabolism and usually leads to cells damage. ROS can be removed by glutathione and may be targeted by *Let-7* as predicted (Table 3). Increases in glutathione peroxidases (GPX) are observed in acute myeloid leukemia, including GPX1, GPX4, and GPX7. GPX1 and GPX4 have been demonstrated to be decreased by *miR-202* to regulate mitochondria stability [153].

### 4.8. Mitochondria Stability

Additionally, mitochondria stability-related proteins, such as monoamine oxidase A, may be linked to *Let-7* as well. In gastric cancer, monoamine oxidase A is correlated with poor prognosis. Knockdown of monoamine oxidase A suppresses mitochondria respiration and glycolysis [154]. Similar results can be observed in lung cancer. Monoamine oxidase A has a positive to poor prognosis and can promote aerobic glycolysis through HK2 [155]. In prostate cancer, increased monoamine oxidase A induces ROS-mediated apoptosis under androgen deprivation [156].

## 5. Conclusions

Even though *Let-7* was the first miRNA identified, its related biological functions linked to diverse biological processes, including glycolysis and autophagy, remain obscure. In this review, we performed a literature review and omics data analysis to generate simulated results to elucidate how *Let-7*-mediated autophagy participates in glucose metabolism, revealing possible molecules that may participate in this regulatory network. However, the related processes may differ from different genetic backgrounds, cancer types, and therapeutic strategies. Mainly, nutrient uptake is the primary means of maintaining fundamental cellular functions. Directly targeting the *Let-7* family to improve the imbalance in nutrient metabolism in cancer cells and minimize the side effects caused by intense treatment is challenging. Therefore, we provide a comprehensive review and detail the regulations and connections between the *Let-7*-family-related glycolysis and autophagy in cancer progression.

## Figures and Tables

**Figure 1 ijms-23-00113-f001:**
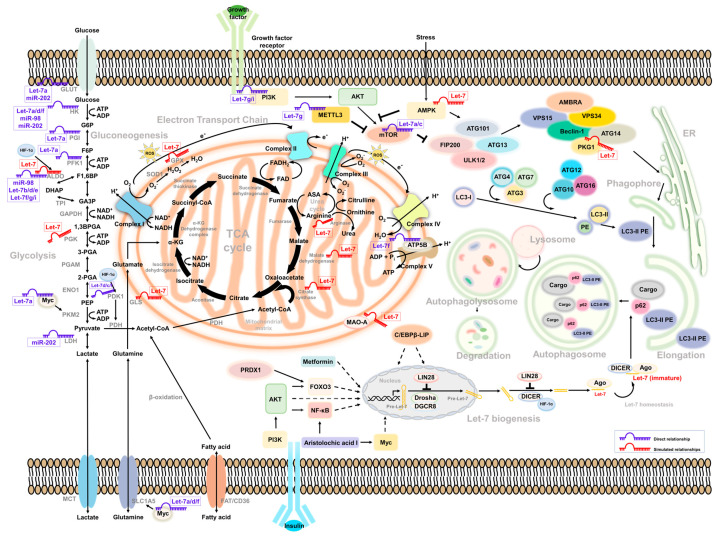
Intermediate mediators/molecules between Let-7-associated glucose metabolism and autophagy.

**Table 1 ijms-23-00113-t001:** The *Let-7* family in humans.

Let-7 Family	Sequence
Let-7a	UGAGGUAGUAGGUUGUAUAGUU
Let-7b	UGAGGUAGUAGGUUGUGUGGUU
Let-7c	UGAGGUAGUAGGUUGUAUGGUU
Let-7d	AGAGGUAGUAGGUUGCAUAGUU
Let-7e	UGAGGUAGGAGGUUGUAUAGUU
Let-7f	UGAGGUAGUAGAUUGUAUAGUU
Let-7g	UGAGGUAGUAGUUUGUACAGUU
Let-7i	UGAGGUAGUAGUUUGUGCUGUU
miR-98	UGAGGUAGUAAGUUGUAUUGUU
miR-202	AGAGGUAGUAGGGCAUGGGAA

**Table 2 ijms-23-00113-t002:** *Let-7* family in pan-cancer on the basis of literature review to coordinate the related survival correlation between patients with cancer and the *Let-7* family.

Cancer Type	Let-7 Family	Clinical Association	Year	Reference
Acute Myeloid Leukemia	Let-7a	Associated with poor outcome	2013	[16]
	Let-7a-2-3p	Associated with good outcome	2015	[17]
	miR-98	Associated with good outcome	2019	[18]
Breast Cancer	Let-7a	Associated with good outcome	2018	[19]
	Let-7a	Associated with good outcome	2018	[20]
	Let-7a	Associated with good outcome	2019	[21]
	Let-7a	Associated with good outcome	2019	[22]
	Let-7a-5p	Associated with good outcome	2020	[23]
	Let-7b	Associated with good outcome	2018	[19]
	Let-7b	Associated with good outcome	2019	[21]
	Let-7b	Associated with good outcome	2020	[24]
	Let-7b	Associated with good outcome	2020	[25]
	Let-7b	Associated with good outcome	2020	[26]
	Let-7b	Associated with good outcome	2016	[27]
	Let-7c	Associated with good outcome	2016	[27]
	Let-7c	Associated with good outcome	2018	[19]
	Let-7c	Associated with good outcome	2019	[21]
	Let-7c	Associated with poor outcome	2020	[28]
	Let-7d	Associated with good outcome	2018	[19]
	Let-7d	Associated with good outcome	2018	[29]
	Let-7d	Associated with good outcome	2019	[21]
	Let-7e	Associated with good outcome	2018	[19]
	Let-7e	Associated with poor outcome	2019	[21]
	Let-7f	Associated with good outcome	2018	[19]
	Let-7f	Associated with good outcome	2019	[21]
	Let-7g	Associated with good outcome	2011	[30]
	Let-7g	Associated with good outcome	2018	[19]
	Let-7g	Associated with good outcome	2019	[21]
	Let-7i	Associated with good outcome	2008	[31]
	Let-7i	Associated with good outcome	2018	[19]
	Let-7i	Associated with good outcome	2019	[21]
Colon Cancer	Let-7a	Associated with poor outcome	2017	[32]
	Let-7g	Associated with good outcome	2017	[33]
Esophageal Cancer	Let-7b	Associated with good outcome	2012	[34]
	Let-7c	Associated with good outcome	2012	[34]
	Let-7c	Associated with good outcome	2013	[35]
Glioblastoma	Let-7a	Associated with good outcome	2013	[36]
	Let-7c	Associated with good outcome	2021	[37]
	Let-7f	Associated with poor outcome	2018	[38]
	Let-7i	Associated with good outcome	2020	[39]
Liver Cancer	Let-7a	Associated with poor outcome	2018	[40]
	Let-7a	Associated with good outcome	2020	[41]
	Let-7b	Associated with good outcome	2020	[41]
	Let-7b	Associated with good outcome	2020	[42]
	Let-7c	Associated with good outcome	2020	[41]
	miR-202	Associated with good outcome	2020	[43]
Lung Adenocarcinoma	Let-7b	Associated with good outcome	2021	[44]
Melanoma	miR-98	Associated with good outcome	2014	[45]
Mesothelioma	Let-7c	Associated with good outcome	2017	[46]
Ovarian Cancer	Let-7b	Associated with poor outcome	2021	[47]
	Let-7d	Associated with poor outcome	2012	[48]
	Let-7e	Associated with good outcome	2017	[49]
	Let-7f	Associated with good outcome	2013	[50]
	Let-7g	Associated with poor outcome	2016	[51]
	Let-7i	Associated with good outcome	2008	[31]
	miR-98	Associated with good outcome	2021	[52]
	miR-98	Associated with good outcome	2020	[53]
	miR-98	Associated with poor outcome	2019	[54]
	miR-98	Associated with poor outcome	2018	[55]
	miR-202	Associated with good outcome	2020	[56]
	Let-7g	Associated with good outcome	2017	[57]
Pancreatic Cancer	Let-7e	Associated with good outcome	2010	[58]
	miR-202	Associated with good outcome	2021	[59]
Prostate Cancer	Let-7b	Associated with poor outcome	2013	[60]
	Let-7c	Associated with good outcome	2013	[60]

**Table 3 ijms-23-00113-t003:** Simulated results profiling of *Let-7* regulators, linked to autophagy and glucose metabolism. All molecules related to the *Let-7* family were downloaded from the ENCORI database (https://starbase.sysu.edu.cn/index.php, accessed on 20 September 2021) and processed using Ingenuity Pathway Analysis (https://analysis.ingenuity.com, accessed on 20 September 2021) to link to the possible molecular regulations.

	Upstream Regulators						
*Let-7* Regulators	LIN28, AKT, AP1, CREB, E2F1, FOXO1, FOXO3, HIF-1α, Myc, NF-κB						
Reference	[110,111,112,113,114,115,116,117,118,119,120,121,122,123,124,125,126]						
	Downstream Regulators						
*Let-7* family	Glycolysis	TCA cycle	Glutamine	Arginine	Autophagy	Oxidative stress	Mitochondria stability
	ALDOC, PGK1	Citrate synthase, Malate dehydrogenase	Glutaminase	Arginase	AMPK	HIF-1α, Glutathione peroxidases	Monoamine oxidase A
Reference	[68,83,127,128,129,130,131,132,133,134]	[135,136,137]	[61,138,139,140,141,142,143]	[144,145,146,147,148,149]	[150,151,152]	[80,103,153]	[154,155,156]

## Data Availability

To perform the simulated molecular interaction model of the Let-7 family, the related up-regulators and downstream effectors were downloaded from ENCORI database (https://starbase.sysu.edu.cn/index.php, accessed on 20 September 2021), then subjected these molecules to the Ingenuity pathway analysis (IPA) (https://analysis.ingenuity.com, accessed on 20 September 2021) to generate the related graphical summary, canonical pathway, and upstream regulators.

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
