# Peer review of "The Metabolism Reprogramming of microRNA Let-7-Mediated Glycolysis Contributes to Autophagy and Tumor Progression"

_ijms, 2021, doi:10.3390/ijms23010113_

Round 1
Reviewer 1 Report
The review entitled “The metabolism reprogramming of microRNA Let-7-mediated glycolysis contributes to autophagy and tumor progression” which was submitted by Liao et al. summarized the current research linkage of the let-7 network between glycolysis and autophagy, as well as its roles in tumor progression. In general, the main text was well-organized. There were still some concerns which the authors should be think about carefully and some mistakes should be corrected carefully.
- Some mistakes should be corrected carefully, for example, “lat-7” in line 321 should be “let-7”, ”table 2” in line 503 should be “table 3”, “table 1 cont.” in line 86 should be “table 2 cont.” and the “mir-98 and mir-202” (closely after line 86) should be “miR-98 and miR-202” since they indicated different meanings.
- The organization of table 2 should be considered, there were many columns while many of them were not significantly, maybe summarize and condense the important information together and put the whole big table into the supplementary data. The present way showing the information was misleading.
- What is the meaningful for table 3? In my opinion, it was just stacked the predicted molecules together. You should delete the table or present the information condensed using other simple way.
- The two figures (Figure 1 and Figure 2) look very similar, only small parts differently. It’s better to modify and combine them together.
- Some sections look very simple or have not enough evidences, for example, section 4.2 and section 4.3, there were only several sentences or cite only one simple reference. My suggestion is to try to find more related references to convince the readers or combine some sections together to make it more convinced.
Author Response
Referee: 1
The review entitled “The metabolism reprogramming of microRNA Let-7-mediated glycolysis contributes to autophagy and tumor progression” which was submitted by Liao et al. summarized the current research linkage of the let-7 network between glycolysis and autophagy, as well as its roles in tumor progression. In general, the main text was well-organized. There were still some concerns which the authors should be think about carefully and some mistakes should be corrected carefully.
- Ans: We thank the Referee for the time taken reviewing our work and for the constructive comments.
1) Some mistakes should be corrected carefully, for example, “lat-7” in line 321 should be “let-7”, ”table 2” in line 503 should be “table 3”, “table 1 cont.” in line 86 should be “table 2 cont.” and the “mir-98 and mir-202” (closely after line 86) should be“miR-98 and miR-202” since they indicated different meanings.
- Ans: We thank the Referee for bringing up this important point and agree with the Referee’s corrections. We apologize for these mistakes and have modified the related descriptions. As requested by the Referee, we have corrected spelling and replaced tables properly as indicated below:
Please refer to the line 324, “lat-7” has been corrected to “let-7”.
Please refer to the line 373, “Table 2” corrected to “Table 3”.
Please refer to lines 84-88, the label mistakes of “table 1 cont” has been replaced by the new edited “Table 2” and we changed “mir-98 and mir-202” to “miR-98 and miR-202” that refers to the mature form of miRNA throughout the manuscript.
2) The organization of table 2 should be considered, there were many columns while many of them were not significantly, maybe summarize and condense the important information together and put the whole big table into the supplementary data. The present way showing the information was misleading.
- Ans: We apologize for the misleading phrasing and agreed with the Referee’s comments. We have modified the Table 2 as requested by the Referee. Table 2 has been re-arranged and extended to better connect the prognosis relationship between Let-7 family and pan-cancer. ​In the replaced Table 2, accordant and updated references relevant to the Let-7-associated tumorous outcome have been included and re-organized according to the related cancer types.
Please refer to lines 86-88, Table 2.
3) What is the meaningful for table 3? In my opinion, it was just stacked the predicted molecules together. You should delete the table or present the information condensed using other simple way.
- Ans: We thank the Referee for pointing out the importance. We agree with the Referee’s comments. As requested by the Referee, we have presented the information condensed as possible. To fit more precisely with our section 4 "The possible connections between let-7-mediated glycolysis and autophagy in cancer progression", in the replaced Table 3, we have included the molecules and pathways from our simulated results adopted via IPA analysis, specifically highlighting important correlations that have been described accordant to literature review.
Please refer to lines 373-378, Table 3.
4) The two figures (Figure 1 and Figure 2) look very similar, only small parts differently. It’s better to modify and combine them together.
- Ans: We appreciate the Referee for raising the valuable questions. We have now included the modified and combined images as Figure 1. Additionally, the direct and indirect relationships between Let-7 family mediated glycolysis and autophagy are highlighted with blue and red color, respectively. More details of Let-7-associated interaction between glycolysis and autophagy processes are given in the figure legend.
Please refer to lines 89-95, Figure 1
5) Some sections look very simple or have not enough evidences, for example, section 4.2 and section 4.3, there were only several sentences or cite only one simple reference. My suggestion is to try to find more related references to convince the readers or combine some sections together to make it more convinced.
Ans: We thank this Referee for their constructive suggestion. We agree with the Referee’s comments. As requested by the Referee, we have found more related references and combined Subsections 4.2, 4.3 and 4.6. For the integrated section, we reorganized the related studies where possible. We have also included numerous descriptions based on our simulated results.
Please refer to lines 436-471, Sections 4.2 Glycolysis, 4.3 TCA cycle, and lines 519-532, 4.7 Oxidative stress. The description of “PGK1 has been observed to be correlated with poor prognosis of glioblastoma [127] and has been demonstrated to bind directly to the Beclin1 and ATG14, two ATGs required for the autophagy process [127,128]” was added in lines 442-445. Another description regarding Let-7 and PKG1 “Another similar postulation can be raised for Let-7 and PKG1. PGK1 and HIF-1α can form a regulatory circuit for controlling breast cancer metastasis [131]. Additionally, Myc serves as an upstream regulator of Let-7 (Table 3), while PGK1 is a downstream target of Myc [132-134]. the Let-7 might function as a linker for Myc to carry out post-transcriptional modification on PGK1, given the possible connection between PKG1 and Let-7 [83].” was also added in lines 452-457. The description explaining the link between Let-7 and TCA cycle “However, how PGK1, citrate synthase, and malate dehydrogenase are regulated by Let-7 has not been fully investigated. The simulated results are provided, and the potent Let-7 downstream molecules including citrate synthase and malate dehydrogenase that may be involved in autophagy regulation are documented in Table 3.” was added in line 466-469. We have also combined the original Subsections 4.7.1 and 4.7.2 as “4.7 Oxidative stress” and added the sentence “The production of excess reactive oxygen species (ROS) is harmful for cells metabolism and usually leads to cells damage.” in lines 527-528.
We thank the Referee for this important observation and related comments as well as professional Referee’s work on our manuscript. We hope that our revised manuscript is acceptable for publication in International Journal of Molecular Sciences.

Reviewer 2 Report
The review focused on microRNA Let-7 network and possible linkage between glycolysis and autophagy, and its role in tumor progression. There have been several studies reported the association of Let-7 miRNA family in multiple cellular and biological functions, including glucose metabolism and autophagy. Therefore, this review is a valuable addition in this field and also provide extensive knowledge for developing alternative cancer treatment strategies by the regulation of cellular metabolism.
Some minor spelling and grammar checks needs to be done.
Author should provide more descriptive figure legend for images.
More up to date scientific literature should be included in the manuscript related to the field as is limited and this study will cover diverse scientific community.
Author Response
Referee 2:
The review focused on microRNA Let-7 network and possible linkage between glycolysis and autophagy, and its role in tumor progression. There have been several studies reported the association of Let-7 miRNA family in multiple cellular and biological functions, including glucose metabolism and autophagy. Therefore, this review is a valuable addition in this field and also provide extensive knowledge for developing alternative cancer treatment strategies by the regulation of cellular metabolism.
- Ans: We deeply appreciate this Referee for their positive and insightful suggestions.
1) Some minor spelling and grammar checks needs to be done.
- Ans: We thank the Referee for pointing this out to us. To address the Referee's concern, we have corrected several spelling mistakes throughout the manuscript and sent the paper to the MDPI English editing service for further spelling and grammar proofreading to improve English quality.
Please refer to the previous English certificate (ID: 36301) and new English certificate (ID: 38305) of this revision.
2) Author should provide more descriptive figure legend for images.
- Ans: Thanks to the Referee for raising this comment. We agree with the Referee that figure legend should provide a more descriptive explanation. Considering reviewer 1’s comments, Figures 1 and 2 have been merged. We have also added a more informative description to explain the possible roles of the Let-7 family in the regulation of glycolysis and autophagy and clarify the molecules/mediators that can be regulated by Let-7 in its miRNA modification. Additionally, the direct and indirect relationships between Let-7 family mediated glycolysis and autophagy are highlighted with blue and red color, respectively.
Please refer to lines 91-95, Figure 1. A more detailed description “The diagram summarizes the current participation of the Let-7 family in the regulation of glucose metabolism and autophagy. The direct (marked blue) and indirect (marked red) interrelationship between glycolysis- and autophagy-related pathway were highlighted according to the simulated model. Molecules and factors involves in the biogenesis of Let-7 and the speculation of its interaction with glucose metabolism and autophagic degradation were also illustrated. Possible molecules that regulate the Let-7 homeostasis in between non-carbohydrate metabolism and autophagy processes were indicated.” in the figure legend was added.
3) More up to date scientific literature should be included in the manuscript related to the field as is limited and this study will cover diverse scientific community.
- Ans: We thank the Referee for this comment, we agree with the Referee that more up to date scientific literature should be included in the manuscript. According to the current understanding, specific studies that emphasize on the interrelationship between Let-7-mediated glycolysis and autophagy, as well as its regulations on tumor progression are limited. Nevertheless, the most updated references associated with Let-7-related clinical outcome in pan-cancer are listed in Table1. Moreover, updated references associated with regulators that linked to autophagy and glucose metabolism were listed in Table3. The updated references that investigated possible molecules/mediators between Let-7-related glycolysis or autophagy pathway were included in section 4 to strengthen the connections between let-7-mediated glycolysis and autophagy in cancer progression.
Please refer to lines 86-88, Table 2. Let-7 related references were provided according to diverse cancer types.
Please refer to lines 373-378, Table 3. Related references of Let-7 family related regulators that link glycolysis and autophagy process were provided.
Please refer to lines 436-457, Section 4.2 Glycolysis. Updated references [127,128] providing more information of Let-7-related regulation in glycolysis and autophagy were added in the sentence “PGK1 has been observed to be correlated with poor prognosis of glioblastoma [127] and has been demonstrated to bind directly to the Beclin1 and ATG14, two ATGs required for the autophagy process [127,128]”.
We thank the Referee for this important observation and related comments as well as professional Referee work on our manuscript. We hope that our revised manuscript is acceptable for publication in International Journal of Molecular Sciences.

Round 2
Reviewer 1 Report
The authors performed the revision carefully and I had no more concerns.